# Research on the Effect of Digital Economy on Carbon Emissions under the Background of “Double Carbon”

**DOI:** 10.3390/ijerph20064931

**Published:** 2023-03-10

**Authors:** Sainan Cheng, Guohua Qu

**Affiliations:** 1School of Business Administration, Shanxi University of Finance and Economics, Taiyuan 030006, China; 2School of Management Science and Engineering, Shanxi University of Finance and Economics, Taiyuan 030006, China

**Keywords:** digital economy, carbon emissions, broadband China, green technological innovation, environmental regulation, heterogeneousness

## Abstract

(1) Background: In light of the global economy’s digitalization and the “double carbon” target constraint, the digital economy is essential to fostering scientific and technological innovation, green growth, and lowering energy emissions. (2) Methods: This paper measures the digital economic index and carbon emission intensity and analyzes their characteristics in spatial and temporal dimensions using 282 Chinese urban panel data by improving various statistical methods of panel data, such as the entropy method, fixed effect model, multi-period DID model, moderating effect model and intermediary effect model. This paper examines the extent and mechanism of the digital economy’s impact on urban carbon emissions. (3) Results: During the sample period, the overall trend of the digital economy in China was one of constant growth, showing an unbalanced distribution pattern of “high in the eastern regions, lower in the central regions and lowest in the western regions” in the spatial dimension. Carbon emissions can be significantly decreased by the digital economy, which has a dynamic effect and an inverted U-shaped trend in its influence. The digital economy plays a significant role in reducing carbon emissions through the rational layout of industrial structures. The transmission mechanisms for the digital economy’s goal of reducing carbon emissions include environmental regulation and green technology innovation. (4) Conclusion: The research findings provide a reference for multiple decision makers to better formulate carbon emission policies and realize carbon emission decrease in the digital economy.

## 1. Introduction

The increase in carbon emissions brought about by industrialization has wreaked havoc on the ecological environment, and the greenhouse effect as well as atmospheric pollution threaten human life and health [1,2]. Increasing demand for fossil energy sources also hastens environmental climate change [3,4], which is becoming a huge challenge and obstacle to the sustainability of society [5]. The growth of GDP is always accompanied by the growth of carbon emissions with China’s modernization. China owns the largest energy consumption system in the world, of which fossil energy accounts for more than 80%. The “double carbon” goal requires that non-fossil energy significantly replace fossil energy in the next 30 years. The matching of clean energy development and energy demands has become an important means to intervene in the peak of carbon emissions. As China’s industrialization progresses, the problems of low-end manufacturing, high environmental pollution and high energy consumption, which have emerged one after another and become increasingly prominent, should not be underestimated. Ten years after China’s accession to the WTO, the carbon emission intensity and per capita carbon emission of cities in Chinese cities displayed a single peak distribution, but a local steady state will appear in the future [6]. Under the “double carbon” goal constraint, controlling the impact of economic activities on the consumption of resources and the environment, while discussing steady economic development needs to be conducted urgently.

At present, the transformation of economic momentum depends more and more on the digital economy, and also occupies a place in promoting economic sustainability. Digitalization is a critical driver for China’s economic transformation and development, and greening is a great aim for China’s high-quality growth. The digital economy will alleviate the imbalance of regional development, promote the formation of a new pattern of regional innovation and coordinated development, and become an important driving factor of economic development, quality change, performance change and power change. It is urgent to study whether the reduction of urban carbon emissions is affected by the digital economy, how the mechanism is transmitted, and how to construct and form a new situation of urban green development of “digital support, innovation linkage, regional synergy, green growth, energy conservation and emission reduction”.

There are two major points of view pulling against each other about the relationship between the digital economy and carbon emissions. The first is the potential reduction in carbon emissions brought on by the digital economy. The advancement and applications such as the Internet, big data and cloud computing are the methods for China to effectively reach its carbon neutrality goal [7]. Digital government facilitates the market to better stimulate the digital economy’s growth potential, thus in turn empowering green technological innovation [8]. Urban green technology innovation has greatly benefited from the growth of the digital economy, which has also helped to lower urban carbon emissions. [9]. The digital economy can not only affect carbon emission intensity directly but also indirectly by optimizing the industrial structure and stimulating scientific and technological innovation [10]. Second, the digital economy is not conducive to reducing carbon emissions. The economic division of labor and its resulting resource locking and development path dependence, combined with insufficient incentives for resource and environmental efficiency, has led to a gradual disadvantage in green growth in resource-rich cities and has created a “curse” on local green economic growth [11]. Telecommunications infrastructure promotes more low-quality technological innovations without positively influencing high-quality green technological innovations [12].

The digital economy now serves as a new source of high-caliber development, and in the context of peak carbon dioxide emissions and carbon-neutral strategy, it is critical to understand what kind of impact the digital economy will have on energy emissions and how it will take effect. Using data from 282 Chinese cities from 2011 to 2019, this article experimentally examines the ecological effects of the digital economy on China’s “double carbon” target constraint. It provides localized suggestions for China to formulate the double carbon policy, as well as a reference for other developing countries to conceive the peak carbon dioxide emissions plan. Firstly, we theoretically propose the direct impact path of the digital economy affecting energy consumption intensity, and further, we clarify the mechanism of the digital economy affecting urban carbon emission from the channels of industrial structure, environmental regulation, and green technology innovation. It also expands and enriches the theoretical research scope of energy saving and emission reduction effects of the digital economy. This paper empirically tests the multidimensional theoretical hypotheses by constructing a series of models and corroborates the ideas in this paper. Under the “double carbon” goal, this paper provides a feasible theoretical basis and policy guidance on how to guide all stakeholders to take rational policy measures, promote the balanced development of the regional digital economy, and give full play to the ecological effects of the digital economy.

## 2. Literature Review

### 2.1. Digital Economy

The digital economy is a result of the Internet’s promotion of the rapid advancement of information and communication technology (ICT). Digital technology and a series of related applications have produced a disruptive reshaping effect on the field of human activities, spreading faster than any previous technological innovation, and the digital economy is growing particularly rapidly in developing countries. The term “digital economy” refers to a portion of the economy that drives its entire or major source from digital technology and has a business strategy centered around digital goods and services [13].

The digital economy owns the ability to foster high-quality economic development, and it also has an essential function in environmental protection and sustainable development and contributes to alleviating air pollution [14]. The growth of the digital economy in the surrounding regions can restrain local environmental pollution, and the digital economy can restrain environmental pollution via the green development effect and innovative development effect. This “industrial pollution emission reduction effect” is primarily attributable to the increase in energy efficiency brought about by technological progress bias and the clean transition of industrial manufacturing mode. The digital economy can decrease industrial pollution without affecting industrial output. The digital economy has also benefited social governance mechanisms and social transformation. A sustainable digital economy has confirmed several intermediate roles between the digital economy and the social governance mechanism, providing a rationale for the government to develop a strong social governance mechanism by completing the digital economy [15]. The digital economy improved the effectiveness of public health care during the presence of COVID-19 by enhancing government performance and regulatory quality [16].

Similarly, through the spatial spillover effect, the digital economy may both directly influence and promote green growth in neighbouring areas [17]. Green development is spatially heterogeneous, and industrial structure and technology innovation are essential channels for green growth in the digital economy [18]. From the perspective of neoclassical economics, the digital economy facilitates the improvement of transaction efficiency and the evolution of the division of labor mode through technological change, but it does not see the digital economy’s influence on green development as being no single or linear [19]. By increasing the degree of economic openness, optimizing the industrial structure and expanding the market potential, the level of green innovation can be indirectly increased by the growth of the digital economy [20]. As an emerging economic form, the digital economy can support both sustainable and high-quality economic development through scientific and technological innovation, technological progress, knowledge spillovers, industrial upgrading, etc. By stimulating the development of renewable energy, it can alleviate the consumption of fossil energy and promote the government’s low-carbon governance and green economic development.

### 2.2. Carbon Emissions

The United Nations’ Kyoto Protocol calls for a 5.2% decrease in greenhouse gas emissions from 1990 to 2012. The greenhouse gases mentioned mainly include CO_2_, CH_4_, N_2_O, PFCs, HFCs and SF_6_ [21]. The Paris Agreement’s long-term goal is to keep the rise in average world temperature to 1.5–2 °C below what it was throughout the industrial era [22]. At the 75th UN General Assembly, China publicly presented the double carbon goal, which aims to reach the peak of carbon emissions by 2030 and carbon neutrality by 2060 [23]. The “Blue Book of Low-Carbon Development: Report on The Carbon Neutrality (2022)” points out that during the period from 2001 to 2010, China’s carbon emissions maintained the same high growth rate as its GDP, once reaching as high as 18%. Global warming is significantly impacted by traffic-related greenhouse gas emissions, which are increasing by around 16% per year in China [24]. According to the Organization for Economic Co-operation and Development (OECD), the average PM2.5 concentration in China in 2019 was 24.6% lower than that in 2013, demonstrating that China has made great efforts and considerable contributions to reducing carbon emissions [25].

Accelerating the green transformation can aid in mitigating the greenhouse effect. Strengthening energy technology innovation and accelerating industrial transformation can better restrain carbon emissions [26]. Green finance can improve the efficiency of carbon emission reduction and significantly reduce carbon emissions locally and nearby, but the influence of green finance on the spatial spillover effect of carbon emission reduction efficiency in surrounding areas may not persist for a long time [27]. The green credit impact and the carbon trading effect are two subgroups of the carbon financing effect, and the carbon emission reduction efficiency can be improved by the gradually expanding carbon financial trading and market mechanism of carbon finance to solve the carbon emission problem [28]. Energy conservation and emission reduction are the most economical and direct paths to green and sustainable development. Optimizing carbon emission transfer is critical for promoting economic expansion and reducing carbon emissions.

### 2.3. The Digital Economy and Carbon Emissions

The impact effects, and transmission mechanisms, of the connection between the digital economy and carbon emissions, are summarized.

The first is the effect the digital economy has on carbon emissions directly. The digital economy development has become one of the main engines to promote economic growth, moreover, sustainable and low-carbon urban growth can be efficiently supported by the digital economy. Regions with a well-developed digital economy have higher carbon emission efficiency, and the digital economy is beneficial for stimulating industrial energy conservation and emission reduction [29]. In studies on direct and indirect structural effects, a considerable negative impact on the assumed carbon emissions is caused by the production structure factors related to the digital industry in China from 2002 to 2017 [30]. Digital finance can significantly reduce carbon emissions at the national level [31]. The measurement results of the digital global value chain show that the influence of the digital global value chain on emission reduction is remarkable. The rise of carbon emissions is inhibited because the negative population effect outweighs the benefits of economic growth and rebound. [32]. Energy usage in middle-income and high-income nations is significantly moderated by digitization [33]. Although increasing energy efficiency can reduce carbon emissions, the digital economy’s expansion is unfavorable for achieving this, which indirectly results in higher carbon emissions [34].

The second is the digital economy’s transmission path regarding the reduction of carbon emissions. The main routes through which the digital economy influences low-carbon growth include environmental governance, technical innovation, industrial structure upgrading, and so on [35]. The digital economy can influence the manufacturing industries to undergo a green and low-carbon transformation through enhanced technological innovation. Developing digital product trade also helps to reduce carbon emissions through technological effects. Carbon emissions are inhibited through innovation and industrial structure upgrading effects and show significant spatial spillover effects. Comprehensive development of the digital economy can restrain regional carbon emissions through industrial progress and energy consumption optimization [36]. It is found that the centrality of the network has a favorable regulatory impact on this mechanism and that the digital economy may improve the decoupling of carbon emissions by optimizing industrial structure [37]. Data from 30 Chinese provinces between 2013 and 2020 were examined, and it was shown that digital financial inclusion can minimize CO_2_ emissions by decreasing per capita energy consumption and increasing per capita GDP [38]. Sub-indicators of the digital economy, such as innovative applications, economic growth, infrastructure and employment, also have a negative influence on carbon emissions. Through increasing green technology innovation, decreasing the amount of coal use, and growing the tertiary industry’s economy, carbon emissions can be indirectly reduced [39].

The digital economy can play a direct role in carbon reduction, and this role frequently has a spatial effect. The digital economy may potentially influence carbon emissions through scientific and technological progress, green technological innovation, industrial structure optimization, environmental regulation, financial development, foreign trade, energy consumption and other aspects.

In this paper, greater attention ought to focus on the extent of the effect and influence of the digital economy on local carbon emissions. An essential component of high-quality economic development is reduced energy emissions, and the digital economy can reach organic integration of green and innovative development. There has not been a wealth of studies to examine and empirically test the mechanisms of how the digital economy decreases the intensity of urban carbon emissions, specifically in the domains of government environmental regulation and enterprise green technology innovation. The paper focuses on how the urban carbon emissions of the digital economy are affected by the issues of the high energy consumption of the initial infrastructure, the digital economy’s uneven development, insufficient power of industrial structure transformation, and the slow development of green technologies that have emerged one after another in the digitally driven high-quality economic growth. The major contributions are: (1) We empirically examine the positive effects that the digital economy has in reducing urban carbon emissions by using Chinese prefecture-level city data as the research sample providing a reference from the field of the digital economy for developing carbon emission schemes for Chinese cities in different regions and of larger population sizes. (2) In contrast to the existing simple static studies on the relationship between the digital economy and carbon emissions, the dynamic effects of the digital economy on urban carbon emissions are examined from the perspective of temporal dynamics. (3) It not only theoretically analyzes the magnitude and direction of the digital economy’s direct impact on carbon emissions but also reveals the intensity of the digital economy’s impact on carbon emissions at the industrial restructuring level, as well as the transmission mechanism played by environmental regulation and green technology innovation, expanding and enhancing the analysis of the theoretical mechanism behind the reduction in carbon emissions caused by the digital economy. It provides the government with an essential theoretical foundation on which to make rational digital economy development strategies and green innovation decisions.

## 3. Research Hypothesis

### 3.1. Direct Impact of the Digital Economy on Carbon Emissions

The environmental effects of the digital economy can be studied from three aspects: macro-development, meso-industry and micro-enterprise. Data and information technologies serve as the foundation of the digital economy. Under the macro backdrop of digital transformation, traditional production relationships and lifestyles have undergone significant upheaval as a result of the digital economy, offering favorable conditions for breakthrough innovation [40]. Digital technologies with high efficiency, low cost and less resource loss, a fully laid out digital infrastructure, and an innovative and inclusive digital transformation environment have significantly promoted the speed of technological innovation, technology diffusion efficiency, and enterprise production efficiency, not only reducing urban carbon emissions, but also significantly reducing industrial dust and wastewater emissions, and positively and continuously affecting high-quality development [41]. From the intermediate perspective, the manufacturing industry in China has remained in the low-side segment for a long time, with a high proportion of resource-consuming industries. Relying on digital technology to improve productivity and encourage the industry’s transition from low-tech to high-tech, thus accomplishing the integration of the real economy and the digital economy to accelerate industry transformation. By vigorously cultivating strategic emerging industries, we will promote the transformation of industrial structure from factor input-driven low-end industries to innovation and data factor-driven ones. Moreover, industrial structure optimization and technological advancement will be fed back to green technological innovation, thus effectively alleviating environmental pollution and improving air quality [42]. At the micro level, with the gradual promotion of the government’s “double carbon” goal, enterprises begin to develop and implement technological innovations with low resource consumption conditions in the process of consolidating green innovation infrastructure [43]. The e-commerce platform mode of customer participation also provides enterprises with more possibilities for development and innovation in the manufacturing processes, which helps to reduce the energy consumption of logistics and transportation [44]. The digital economy can effectively promote enterprises to rationally allocate production resources, encourage clean energy application and development, innovate green production and manufacturing modes, and constrain enterprises to undertake social responsibilities. Enterprises essentially take the initiative to adopt green production and reduce energy consumption intensity and carbon emissions. The impact of the enterprise’s digital transformation on carbon reduction is reflected in the key production and operation links of the digitalized enabling enterprises, promoting green production and intelligent services, and promoting enterprises to become green and low-carbon. Promote the digital transformation of enterprises from key processes through the layout of digital management systems. For example, the dynamic monitoring of the warehouse is conducive to the rational planning of inventory, reducing the labor cost of warehouse management, improving the management efficiency of products and materials, and promoting further cost reduction and efficiency increase of enterprises. Therefore, the article puts forward the following hypothesis.

**Hypothesis** **1** **(H1).**
*The digital economy helps to reduce urban carbon emissions.*


### 3.2. The Inverted U-Shaped Influence Trend of the Digital Economy on Carbon Emissions

The digital economy can assist in enhancing the aggregation and function mode of elements by increasing the circulation speed of data elements, the innovation speed of technology, and the scope of technology spillovers. As a result, enterprises may devote more labor and material resources to technological R&D and innovation, increase enterprise production efficiency, and increase energy conservation and emission reduction [45]. Economic agglomeration has a facilitating and then inhibiting effect on green economic efficiency, which is transmitted through infrastructure, advanced level of the labor market and environmental regulation [46]. As opposed to that, the development level and innovation basis of the digital economy vary greatly among China’s four regions. The innovation effect of digital economy development in cities above the national average level is weaker than that in cities with a low growth degree of the digital economy [47]. In the early stages of regional digital economy growth, the main focus is on infrastructure construction and layout, and the effect on innovation is mainly in the low-tech aspects. The digital economy will have a qualitative change for innovation with the gradual enrichment of a series of digital-related economic activities. Both highly and less developed areas of the digital economy are experiencing unsynchronized changes in carbon emissions and the digital economy [48]. Moreover, green technological innovation shows obvious nonlinear characteristics of increasing marginal effect in the process of the digital economy affecting industrial structure optimization [49]. It implies that the digital economy, urban innovation, and carbon emission may not be correlated linearly. Therefore, the article proposed the following hypothesis.

**Hypothesis** **2** **(H2).**
*An “inverted U” trend may exist in the impact of the digital economy on urban carbon emissions.*


### 3.3. The Regulatory Role of Industrial Structure in the Process of the Digital Economy Affecting Carbon Emissions

The digital economy has subverted the traditional closed innovation environment and industrial development structure, becoming a key driving force in promoting green innovation, industrial structure adjustment and high-quality development, as well as a future direction for reducing fossil energy consumption, breaking through resource and environmental constraints, and seeking green economic growth. Environmental target constraints will make local governments enhance the transformation and upgrading of local industries by tightening environmental regulations and adjusting industrial policies and fiscal expenditure structure [50]. The digital economy’s development can encourage the rationalization and upgrading of industrial structure, enhance the quantity and quality of urban green innovation, and thus reduce the intensity of regional carbon emissions [51], and this effect is regionally heterogeneous. Eastern China is more affected negatively by the growth of the digital economy in terms of carbon emissions, and the areas located within urban agglomerations are more affected by the digital economy [52]. Digital industrialization, as the foundation of digital economy development, continues to infiltrate and merge with traditional industries, leading to a dynamic process of industrial structure upgrading and optimization gradually [49]. Thus, the following hypothesis is proposed.

**Hypothesis** **3** **(H3).**
*Industrial structure plays a regulatory role in the process of the digital economy affecting carbon emissions.*


### 3.4. Transmission Ways of Digital Economy Affecting Carbon Emissions

Low-carbon technology innovation is the primary motivator for increasing carbon total factor productivity [53]. The digital economy is progressively becoming a major driving force of development represented by regional low-carbon. The three primary ways that the digital economy influences LCD are through governance, technological advancement, and industrial upgrading improvement [35]. The digital economy can significantly empower urban innovation ability, thus exerting the innovative carbon emission reduction effect. The urban innovation environment is transformed into the productivity of enterprises by attracting talent. Effective sharing and application of knowledge management can improve the innovation performance of enterprises [54]. The inclusive development of digital finance under the digital economy provides convenient financing support for enterprises’ green technology innovation, and urban wealth can play a role in attracting and screening innovative technologies, prompting green technology agglomeration and evolution towards green development [55]. Thus, we propose Hypothesis H4.

**Hypothesis** **4** **(H4).**
*The digital economy reduces carbon emissions by promoting green technological innovation.*


It is mutually reinforcing to play the role of effective government and market in the course of generating environmental benefits in the digital economy. The government controls the green production of enterprises and industrial chains through policy constraints, financial support, tax control, land transfer and other means, and restrains the negative externalities such as water and air pollution emissions generated by enterprises, to curb the total carbon emissions in the region. Tax incentives may compensate enterprises for the cost of green technology acquisition, but may squeeze out enterprises’ green innovation capability. Moreover, while enterprises comply with the government’s environmental protection requirements, it may have the consequence of increasing the number of green patent applications in the short term and the poor quality of green technology innovation in the long term. The function of low-carbon city pilot policies in inducing green technology innovation at the overall enterprise level is mostly performed through required policy instruments [56]. Hypothesis H5 is proposed consequently.

**Hypothesis** **5** **(H5).**
*The digital economy affects carbon emissions through environmental regulation.*


Based on the theoretical mechanism analysis, a research framework was drawn, as shown in Figure 1.

## 4. Research Design

### 4.1. Model Setting

#### 4.1.1. Fixed Effect Model of the Digital Economy Affecting Urban Carbon Emissions

The impact of the digital economy on urban carbon emissions is measured using a fixed effect model, which is in Formula (1) [54], where *i* indicates different cities, *t* means different years, and *COE*_*i*,*t*_ refers to urban carbon emission, *DIGi*,*t* represents the urban digital economy, *CONTROL*_*i*,*t*_ incorporates several control variables, including *FAS*, *IND*, *GOV*, *EDU*, *MAR*, *OPEN*. The term *μ_i_* represents individual effects, *δ_t_* indicates time-fixed effects and *ε*_*i*,*t*_ means random disturbance terms. The term “*β*_0_” is the regression’s constant term in Formula (1), “*β*_1_” indicates the regression coefficient of *DIGi*,*t* to *COE*_*i*,*t*_, and the regression coefficient of *CONTROL*_*i*,*t*_ to *COE*_*i*,*t*_ is denoted by “*β*_2_”.
*COE*_*i*,*t*_ = *β*_0_ + *β*_1_*DIG*_*i*,*t*_ + Σ*β*_2_*CONTROL*_*i*,*t*_ + *μ_i_* + *δ_t_* + *ε*_*i*,*t*_
(1)


#### 4.1.2. Multi-Period DID Model of the Digital Economy Affecting Urban Carbon Emissions

Formula (2) shows the multi-period DID model. *BRC*_*i*,*t*_ represents the processing period dummy variable that varies by individual. If individual *i* is treated in period *t*, representing entry into the treatment period, then all values thereafter are 1; otherwise, the value is 0 [57]. In Formula (2), “*β*_0_” is the constant term of the regression, the regression coefficient of *BRC*_*i*,*t*_ to *COE*_*i*,*t*_ is denoted by “*β*_1_”, and “*β*_2_” is the regression coefficient of *CONTROL*_*i*,*t*_ to *COE*_*i*,*t*_.
*COE*_*i*,*t*_ = *β*_0_ + *β*_1_
*BRC*_*i*,*t*_ + Σ*β*_2_*CONTROL*_*i*,*t*_ + *μ_i_* + *δ_t_* + *ε*_*i*,*t*_
(2)


#### 4.1.3. The Nonlinear Test Model of the Influence of the Digital Economy Development Level on Urban Carbon Emissions

In Formula (3), *DIGPF*_*i*,*t*_ is the quadratic term used to evaluate the digital economy’s level of development to investigate if it has an inverted U-shaped effect trend on carbon emissions [58]. In Formula (3), “*β*_0_” is the constant term of the regression, “*β*_1_” is the regression coefficient of *DIGPF*_*i*,*t*_ to *COE*_*i*,*t*_, and “*β*_2_” is the regression coefficient of *CONTROL*_*i*,*t*_ to *COE*_*i*,*t*_.
*COE*_*i*,*t*_ = *β*_0_ + β_1_*DIGPF*_*i*,*t*_ + Σ*β*_2_*CONTROL*_*i*,*t*_ + *μ_i_* + *δ_t_* + *ε*_*i*,*t*_
(3)


#### 4.1.4. Dynamic Effect Test Model of the Digital Economy Affecting Urban Carbon Emissions

The lag term of the digital economy *DIG*_*i*,(*t*−*n*)_, is created to examine the dynamic impact of the digital economy on carbon emissions. Based on Formula (1), the core explanatory variable’s lag term is added, and *n* represents the lag order [59]. The regression form is shown in Formula (4). In Formula (4), “*β*_0_” is the constant term of the regression, the regression coefficient of *DIG*_*i*,(*t*−*n*)_ to *COE*_*i*,*t*_ is denoted by “*β*_1_”, and the regression coefficient of *CONTROL*_*i*,*t*_ to *COE*_*i*,*t*_ is denoted by “*β*_2_”.
*COE*_*i*,*t*_ = *β*_0_ + *β*_1_*DIG*_*i*,(*t*−*n*)_ + Σ*β*_2_*CONTROL*_*i*,*t*_ + *μ_i_* + *δ_t_* + *ε*_*i*,*t*_
(4)


#### 4.1.5. Moderating Effect Model of the Digital Economy Affecting Urban Carbon Emissions

To investigate how the growth of the digital economy affects urban carbon emissions, the model for the moderating effect is created by adding the interaction term *DIG*_*i*,*t*_*_INR*_*i*,*t*_ between the digital economy and industrial structure rationalization (*INR*_*i*,*t*_) based on Formula (1) [60]. The regression form is shown in Formula (5). The term “*β*_0_” is the constant term of the regression, “*β*_1_” is the regression coefficient of *DIG*_*i*,*t*_ to *COE*_*i*,*t*_, “*β*_2_” is the regression coefficient of *INR*_*i*,*t*_ to *COE*_*i*,*t*_, “*β*_3_” is the regression coefficient of the interaction term *DIG*_*i*,*t*_*_INR*_*i*,*t*_ to *COE*_*i*,*t*_, and *β*_4_” is the regression coefficient of *CONTROL*_*i*,*t*_ to *COE*_*i*,*t*_.
*COE*_*i*,*t*_ = *β*_0_ + *β*_1_*DIG*_*i*,*t*_ + *β*_2_*INR*_*i*,*t*_ + *β*_3_*DIG*_*i*,*t*_*_INR*_*i*,*t*_ + Σ*β*_4_*CONTROL*_*i*,*t*_ + *μ_i_* + *δ_t_* + *ε*_*i*,*t*_
(5)


#### 4.1.6. Intermediary Effect Model of the Digital Economy Affecting Urban Carbon Emissions

To examine the function of innovation in green technology as an intermediator in the process of digital economy development affecting carbon emissions, the models are constructed as Formulas (6) and (7). The regression’s constant term is represented as “*β*_0_” in Formula (6) and (7). In Formula (6), the regression coefficient of *DIG*_*i*,*t*_ to *GTI*_*i*,*t*_ is denoted by “*β*_1_”, and the regression coefficient of *CONTROL*_*i*,*t*_ to *GTI*_*i*,*t*_ is denoted by “*β*_2_”. In Formula (7), the regression coefficient of *DIG*_*i*,*t*_ to *COE*_*i*,*t*_ is represented by “*β*_1_”, the regression coefficient of *GTI*_*i*,*t*_ to *COE*_*i*,*t*_ is expressed by “*β*_2_”, and the regression coefficient of *CONTROL*_*i*,*t*_ to *COE*_*i*,*t*_ is presented by “*β*_3_”.

Similarly, to test the mediating role of environmental regulation, models in Formula (8) and Formula (9) are constructed. Other variables are similar to those in Formula (1) [61]. In Formula (8), “*β*_0_” means the regression’s constant term, “*β*_1_” indicates the regression coefficient of *DIG*_*i*,*t*_ to *ENV*_*i*,*t*_, and the regression coefficient of *CONTROL*_*i*,*t*_ to *ENV*_*i*,*t*_ is denoted by “*β*_2_”. In Formula (9), “*β*_0_” represents the regression’s constant term, the regression coefficient of *DIG*_*i*,*t*_ to *COE*_*i*,*t*_ is expressed by “*β*_1_”, the regression coefficient of *ENV*_*i*,*t*_ to *COE*_*i*,*t*_ is presented by “*β*_2_”, and the regression coefficient of *CONTROL*_*i*,*t*_ to *COE*_*i*,*t*_ is denoted by “*β*_3_”.
*GTI*_*i*,*t*_ = *β*_0_ + *β*_1_*DIG*_*i*,*t*_ + Σ*β*_2_*CONTROL*_*i*,*t*_ + *μ_i_* + *δ_t_* + *ε*_*i*,*t*_
(6)

*COE*_*i*,*t*_ = *β*_0_ + *β*_1_*DIG*_*i*,*t*_ + *β*_2_*GTI*_*i*,*t*_ + Σ*β*_3_*CONTROL*_*i*,*t*_ + *μ_i_* + *δ_t_* + *ε*_*i*,*t*_
(7)

*ENV*_*i*,*t*_ = *β*_0_ + *β*_1_*DIG*_*i*,*t*_ + Σ*β*_2_*CONTROL*_*i*,*t*_ + *μ_i_* + *δ_t_* + *ε*_*i*,*t*_
(8)

*COE*_*i*,*t*_ = *β*_0_ + *β*_1_*DIG*_*i*,*t*_ + *β*_2_*ENV*_*i*,*t*_ + Σ*β*_3_*CONTROL*_*i*,*t*_ + *μ_i_* + *δ_t_* + *ε*_*i*,*t*_
(9)


### 4.2. Variables

#### 4.2.1. Explained Variables

As the statistical caliber of the electricity consumption in the China City Statistical Yearbook changed in 2017 and after, this paper calculates the carbon emissions according to statistical yearbook data. In accordance with the carbon emission coefficient of energy, the carbon emissions produced by the use of electric power, gas, liquefied petroleum gas (LPG), and heat may be converted, and the overall carbon emissions can be obtained by adding them all together [6]. There are two important metrics in carbon emission reduction policy design, one is per capita carbon emissions (the ratio of total urban carbon emissions to total urban population), which is a relatively equity-oriented measurement, and the other is carbon emission intensity (the ratio of carbon emissions to regional GDP), which focuses on the measurement of efficiency [9]. The article uses the carbon emission intensity of cities as the main regression and takes the natural logarithm of carbon emissions for robustness testing.

#### 4.2.2. Core Explanatory Variables

The index system of digital economy composite index (DIG) is established in accordance with the research that has already been conducted [62,63], as shown in Table 1. Weighing the sub-dimensional indicators of the digital economy and calculating the comprehensive index are accomplished using the principal component analysis method and the improved entropy value method.

#### 4.2.3. Adjusting Variables

The positive effect of the digital economy on social and economic activities is also manifested in the promotion of innovation diffusion, technology spillovers and information sharing in a wider range, which has a continuous impact effect. Innovation in digital technology may lower energy use, boost the economy, accomplish energy conservation, and lower emissions, and then reduce carbon emissions by improving the mobility of factor resources in enterprise production, reducing process management costs and improving production efficiency. This paper examines the industrial structure rationalization’s moderating effect (INR). The equilibrium state of the economy is defined as a Theil index of 0, otherwise, it deviates from the equilibrium state [64].

#### 4.2.4. Intermediary Variables

Technology progress will ultimately reduce environmental costs by reducing the carbon emission level, which will in turn enhance the improvement level of urban green innovation. Existing studies measure urban green innovation (GTI) mainly using the number of green patents related to enterprise production. Green technology patents are the number of applications or authorizations [65,66]. Therefore, the article identifies environment-friendly invention patents consistent with green innovation activities using technical information on patented innovation activities provided by the International Patent Classification (IPC) and uses green invention patents to measure the level of green innovation in each city.

There may be two effects of environmental regulation on carbon emissions. First, environmental legislation raises the expense of technical management for businesses, reduces investment in technology advancement, and delays the introduction of human capital, all of which might stifle technological advancement and have no positive impact on lowering carbon emissions. Besides, environmental regulations compel companies to take more social responsibility and take more green technologies and environment-friendly modes of production, thus achieving lower carbon emissions. The article selects industrial wastewater emissions, industrial sulfur dioxide emissions, and industrial smoke (powder) emissions to obtain the environmental regulation composite index (REG) after standardization [67]. The smaller the value, the less pollution emission, the harder the government’s efforts to control environmental pollution, and the stronger the intensity of environmental regulation.

#### 4.2.5. Control Variables

Due to the complexity of the factors affecting urban green innovation, we add a series of control variables to enhance the robustness of the econometric model. The logarithm of capital investment stock represents the total societal investment in fixed assets (FAS) [68]. Industry (IND) is expressed by the ratio of the added value of the second production to GDP. The quality of economic growth is significantly influenced by the macro-control role of local government, and the primary indicators of that function are fiscal revenue and expenditure. The size of the government (GOV) is represented by fiscal revenue as a share of GDP [69]. The improvement of education level (EDU) contributes to the cultivation of talents and the improvement of overall quality. It is vital for the economic development quality to absorb and apply foreign advanced technology, research and develop new technology, and promote regional technological innovation through the construction of a government-industry-academia-research synergistic body. There are various measures of education level, and the article measures human capital by the share of GDP spent on education [58]. The marketization level of an area is fully reflected by the marketization index (MAR). Digital government can “technically empower” national governance, which helps to build a service-oriented government, realize decentralization and function optimization, and provide a favorable business environment and solid institutional guarantee for market players to innovate with green technologies, thus stimulating the vitality of green technology innovation [8]. The data are derived from the Marketization Index of China’s Provinces: NERI Report 2019 [70]. Opening to the outside world (OPEN) can affect the quality of regional economic development through technology spillovers and “pollution shelters” [71]. The ratio of total import and export value to GDP indicates the level of openness. The permanent population of the entire city (10, 000 people) represents the variable population size (POP), and the logarithm of GDP per capita represents the level of economic development (LGDP) [72].

### 4.3. Data

The data are obtained from databases such as China Urban Statistical Yearbook, China Environmental Statistical Yearbook, and China Energy Statistical Yearbook. Individual missing data are supplemented by interpolation. The descriptive statistics for the key variables are displayed in Table 2.

## 5. Results and Discussion

### 5.1. The Digital Economy Index and Spatial and Temporal Distribution Characteristics of Urban Carbon Emissions

Based on the measured data, the distribution of the digital economic index in practice and space is plotted, as shown in Figure 2a which shows the kernel density superposition of China’s carbon emission intensity in 2011, 2013, 2015, 2017 and 2019. It can be seen that the height of the peak shows a decreasing trend year by year, and the peak gradually moves to the right. It shows that while China’s digital carbon emissions are decreasing, the trend of dispersion tends to converge gradually.

Figure 2c depicts the histogram of carbon emission intensity of the capital cities in 2013, 2015, 2017 and 2019. It can be seen that the western regions such as Xining, Yinchuan and Urumqi lead in carbon emissions. The development of these areas mainly depends on abundant natural resources, and resource-consuming industries such as heavy industry, minerals and coal dominate, leading to significantly high carbon emissions. The carbon emissions of Nanning, Hefei and Changsha are at the end and are far lower than in comparable western cities. Overall, the major cities’ carbon emission intensity represents a clear pattern of “high in the west and low in the east”. The eastern, central, and western areas have dramatically different carbon emission intensities but are relatively similar within the regions.

Figure 2b shows the superposition of the kernel density of China’s digital economy development level in 2011, 2013, 2015, 2017 and 2019. The peak may be seen to be trending rightward year after year, and the height of the peak decreases gradually with time. It demonstrates that while China’s digital economy level is continuously developing, its difference is also gradually narrowing. The digital economy’s growth around 2011 was concentrated in a few regions, and progressively evolved into a pattern of regional coordinated development by 2019.

Figure 2d represents the histogram of the development level of the digital economy in China’s capital cities in 2013, 2015, 2017 and 2019. The development level of the digital economy in Guangzhou, Hangzhou, Shanghai and other regions is at the nationally leading level, which is consistent with the empirical facts. These areas are early in developing the digital economy and have policy and geographical advantages such as superior economic foundation, comprehensive digital infrastructure, abundant human capital and policy inclination. The digital economy development level in Chongqing, the gateway to the west, Nanning, Lanzhou and other areas is lower than in other surrounding areas, and there is a fracture. Overall, the development of the digital economy in China’s major cities shows obvious ladder distribution characteristics, with regional agglomeration and relative spatial stability, although with significant differences among regions, the overall differences tend to narrow.

### 5.2. Research on the Impact of Digital Economy on Carbon Emissions

#### 5.2.1. Direct Impact of the Digital Economy on Carbon Emissions

Table 3 reports the direct impact of the digital economy (DIG) on urban carbon emissions (COE). The fixed effect model is chosen using the Housman test. Without using any control variables, Column (1) displays the results of the digital economy’s regression in carbon emissions. The digital economy can effectively reduce carbon emissions and the result is significant at the statistical level of 1%.

Column (2) indicates that the digital economy can substantially lower carbon emissions after adding control variables, and Hypothesis H1 is verified by passing the significant level test of 1%. In terms of economic significance, considering that the average carbon emission intensity is 0.4686, the intensity level of local carbon emission will drop by about 0.3932(0.8392 × 0.4686) on average for each 1% rise in the digital economy’s development level. The possible reasons are that the rapid development of big data technology, based on new digital infrastructure, has promoted the organic integration of traditional factors of production, such as labor capital and data technology, which not only provides rich production materials for enterprises but also contributes to the efficient allocation of innovative factors, thus reducing carbon emissions. What is more, under the background of “double-carbon”, enterprises will tend to promote digital transformation in the direction of environmental protection and energy-saving technology innovation to respond to national policies and avoid environmental punishment and increase green technology R&D investment, which will directly increase the output of innovative activities related to green technology and reduce carbon emissions.

The regression findings in column (2) of Table 3 show that the link between the controllable variables and carbon emissions is consistent with the theory. The coefficient of fixed capital investment stock (FAS) is negative and statistically significant at the 1% level, indicating that fixed capital investment stock helps to reduce carbon emissions. The regression coefficient of industrial structure (IND) is −0.5079, suggesting that industrial structure optimization is beneficial to reduce carbon emissions. The regression coefficient of government size (GOV) is 1.1501. The higher the government revenue, the higher the portion that may be used for fixed investment or municipal construction, which is not conducive to reducing carbon emissions instead. The regression coefficient of education level (EDU) is 5.0974 is significantly positive at the statistical level of 5%, which may be due to the increase in knowledge level and skill accumulation of human capital through higher expenditure on social education. High-education areas are more likely to generate digital economic activities, and overall energy consumption will rise as a result of the resultant digital economy and growth in service demand. The regression coefficient of the marketization level (MAR) is −0.0014, indicating that cities with higher marketization levels have lower carbon emission levels. The coefficient of “opening to the outside world” (OPEN) is 0.2773, indicating that the higher the ratio of GDP’s total import and export value, the richer the conditions for enterprises to realize foreign capital use and technology inflow, the higher the carbon emission intensity.

#### 5.2.2. The Inverted U-Shaped Impact of the Digital Economy on Carbon Emissions

Column (3) of Table 3 reports the influence direction and effect of digital economy development on carbon emissions after adding the digital economy square term (DIGPF). The digital economy’s quadratic term has a regression coefficient of −1.2471, which is significant at the 5% level of statistical significance. The effect of the digital economy on carbon emissions shows an “inverted U-shaped” trend. As the capacity for the digital economy expands, the intensity of carbon emissions rises until it reaches its maximum level during the sample period. The low level of urbanization has made it difficult to lay out the large-scale construction of new infrastructure during China’s industrialization and informationalization processes. The initial stages of construction and use of the digital economy require the assistance of both electric and thermal energy. The construction and maintenance of these resource-intensive infrastructures, as well as the increased energy consumption intensity of the digital economy operation itself, are the major causes. Hence, the architecture and maintenance of digital infrastructure building must consume a lot of energy in the early stages of the growth of the digital economy, raising carbon emissions. As the ability for the digital economy to thrive keeps expanding, once carbon emissions have peaked, their intensity will progressively decline. The carbon reduction effect of the digital economy begins to be brought into play to achieve economies of scale. Hypothesis H2 is verified.

### 5.3. Robustness Test of the Digital Economy Affecting Carbon Emissions

#### 5.3.1. External Impact Test of “Broadband China” Policy

Constructing new infrastructure is an essential component of the digital economy’s transformation of economic kinetic energy. To improve the level and coverage of broadband construction and improve the coordination and progress of urbanization and informatization, the Chinese government has arranged the “broadband China” demonstration cities (urban agglomeration). In three batches in 2014, 2015, and 2016, a total of 120 cities (clusters) were selected as “Broadband China” demonstration sites. The “Broadband China” pilot project is utilized as an endogenous policy impact, and the method of multi-period double difference (DID) is applied for policy evaluation, to more robustly assess whether the digital economy’s growth would result in a reduction in carbon emissions. In urban areas, the layout and construction of broadband infrastructure facilitate the growth of the digital economy, and a good quasi-natural experiment could be provided by the “Broadband China” policy. According to the findings in column (4) of Table 3, the “Broadband China” (BRC) pilot project significantly lowers city-level carbon emissions, all of which are significant at the level of 5%. After adding control variables, the development of the digital economy with the “Broadband China” policy as the carrier has greatly decreased urban carbon emissions, as seen in column (5) of Table 3. The robustness test is passed and it is consistent with the previous conclusion.

#### 5.3.2. Endogenous Test

Carbon emissions can be decreased through the growth of the digital economy, but it is also possible that regions with low carbon emissions are more inclined to develop digital technology, so raising the level of the digital economy’s development. To solve this endogenous issue, the digital economic index lag period (L.DIG) is analyzed. The coefficient value of the digital economic index with a one-period lag is −1.1493, which is significant at 5%, as shown in Column (1) of Table 4, demonstrating that the reduction of carbon emissions is influenced dynamically in the temporal dimension by the development of the digital economy. Therefore, the above regression results are still stable after considering the endogeneity issue. To eliminate the possible interference from the control variables, column (2) of Table 4 also analyzes all the control variables lagged by one period. The robustness of the model is reliable, as shown by the coefficient value of the level of digital economic development, which is −1.2476 for the level of digital economy development and remains significantly positive at the level of 5%and is consistent with the results above. Moreover, the absolute values of the regression coefficients in columns (1) and (2) of Table 4 are greater than 0.8392, indicating that the current period’s digital economy continues to have an inhibitory influence on future carbon emissions, and the effect has been enhanced. This is consistent with the previous conclusion and passes the robustness test.

#### 5.3.3. Changing the Calculation Method of the Development Level of the Digital Economy

The level of development of the digital economy was assessed in the preceding piece of content using the enhanced entropy approach. Further, The robustness test’s digital economic development level (DIG2) is determined using the principal component analysis approach. According to Column (1) of Table 5, the digital economy has a regression coefficient of −0.0392 on carbon emissions, indicating that it can lower carbon emissions. According to Column (2), the quadratic term of the digital economy’s regression coefficient on carbon emissions is −0.0361 and is significant at 1%. It suggests that as the level of development of the digital economy rises carbon emissions exhibit an “inverted U-shaped” trend. The lagged term of the digital economy is taken into account in column (3), and its regression coefficient, which is −0.0676, is significant at 5%. It demonstrates that the inhibition of the digital economy on future carbon emissions continues to be strong. This is consistent with the previous conclusion and passes the robustness test.

#### 5.3.4. Use the Logarithm of Carbon Emissions as the Explained Variable

The measurement index of carbon emission intensity is replaced by the logarithm of carbon emission (COE2) to perform the robustness test. Table 6 displays the results of the regression. The effect coefficient of the growth of the digital economy on carbon emissions is −1.3062, which is displayed in Column (1) and is significant at 1%. In column (2), the digital economy’s quadratic term has a regression coefficient of −2.0543, which is statistically significant at 1%. There is an “inverted U-shaped” accompanying evolution trend between the digital economy and carbon emission intensity. The lagged term for the digital economy is included in the regression model in column (3), and its regression coefficient, which is −1.3560, is significant at the 5% level of statistical significance. It implies that there is substantial temporal heterogeneity in carbon emissions related to the digital economy. The robustness tests support our conclusion.

#### 5.3.5. Eliminate the Robustness Test of Provincial Capitals and Municipalities Directly under the Central Government

The research sample includes 282 Chinese cities with varying economic growth levels, digital economic development abilities and industrial layouts. Considering that the policy tendency of municipalities directly under the central government and provincial capitals is more favorable during the sample period, and the inclusion of these regions in the study may affect the research results, to eliminate such interference, these cities are removed from the sample period to further increase the trustworthiness of the research results. After excluding 30 cities, Table 7’s column (1) demonstrates that the growth of the digital economy significantly reduces carbon emissions. According to Column (2), the early stages of the development of the digital economy and carbon emissions tend to grow together before the reverse tendency emerges. Column (3) shows that the impact of the digital economy on carbon emissions has a lag in the temporal dimension. It passes the robustness test and supports the prior conclusion.

### 5.4. Mechanism Test of Digital Economy Affecting Carbon Emissions

#### 5.4.1. Moderating Effect

The mechanism of green technology innovation concerning the effect of the digital economy on carbon emissions is further examined in this article. Table 8 displays the results from the moderating effect model test of the industrial structure rationalization moderating effect.

The digital economy’s regression coefficient in column (1) of Table 8 is 1.9094, which is significant at 1%. It demonstrates that carbon emissions can be reduced in cities with low levels of industrial rationalization (INR = 0 group) since the digital economy is in the rising stage. It may be that regions with low industrial structure rationalization are still developing their digital economy infrastructure, which results in lower carbon emissions. DIG_INR’s regression coefficient, which is 3.3256, is statistically significant at the 1% level. It suggests that, compared to regions with low and high industrial structure rationalization (INR = 1 group), the digital economy exhibits a smaller influence on carbon emissions.

In areas with low industrial structure rationalization, the digital economy’s regression coefficient to green technology innovation is 1.9094. The regression coefficient of cross-variable DIG_INR is 3.3256, which is the slope difference between regions with higher and lower industrial structure rationalization levels. After the joint test, the regression coefficient in regions with a high level of industrial structure rationalization is 5.235(1.9094 + 3.3256), which is significant at 1%. Its economic significance is that the difference in carbon emission intensity between regions with an extremely high digital economy (DIG = 1 group) and an extremely low (DIG = 0 group) is 2.4531(5.235 × 0.4686) on average, all other conditions being equal. The relationship between the digital economy and carbon emissions is significantly influenced by the rationalization of industrial structure. H3 is verified.

#### 5.4.2. Intermediary Effect

Green technology innovation and environmental regulation are chosen as intermediate variables to examine the intermediary impact to assess the transmission mechanism of the digital economy affecting carbon emissions. Table 9 displays the results of the regression. Taking column (1) as an example, the digital economy significantly reduces urban energy consumption intensity. In column (2), the digital economy’s regression coefficient to green technology innovation (GTI) is 0.6149 and is significant at the statistical level of 0.1, demonstrating that the digital economy has significantly improved the green technology innovation level. The digital economy’s regression coefficient to carbon emissions in column (3) is −0.8582 and is significant at a 1% statistical level. The green technological innovation’s regression coefficient to carbon emissions is 0.0229 and is significant at 10%, demonstrating that the digital economy will reduce the carbon emission intensity with the improvement of the green technology innovation level. Hypothesis H4 is verified.

According to column (4), the digital economy’s regression coefficient to environmental regulation (ENV) is −0.2734 and is significant at the statistical level of 1%, suggesting that as the digital economy develops, the level of environmental control is becoming less intense. In column (5), the digital economy’s regression coefficient to carbon emissions is −0.8354 and is significant at the 5% level. Statistically significant at the 1% level, the regression coefficient of environmental legislation on carbon emissions is 0.5438. It implies that by enhancing the decrease of industrial waste emissions, the digital economy can decrease the intensity of carbon emissions. Hypothesis H5 is verified.

### 5.5. Heterogeneity Analysis of the Impact of the Digital Economy on Carbon Emissions

#### 5.5.1. Regional Heterogeneity

The variety of resource endowments and degrees of economic growth in China will influence the digital economy and the intensity of carbon emissions. As indicated in Table 10, different areas of China are grouped to examine the impact of the digital economy on carbon emissions.

Table 11 displays the findings of subgroup regressions. The impact of the digital economy on carbon emissions in the eastern, central, and western areas are depicted in columns (1), (2) and (3), respectively. The regression coefficients are −0.6258, −1.2342 and −1.3356, respectively, suggesting that various regions have varying effects of the digital economy on lowering the intensity of carbon emission.

#### 5.5.2. Heterogeneity of Urban Population Size

The development of the digital economy in most areas of China is at a low level, and its inadequate and unbalanced development situation is bleak [73]. For regions with high population density, it is essential to seek coordination between economic growth and carbon emission reduction. Theoretically, the population size of different cities may also have an impact on how the digital economy and carbon emissions are related. The samples are divided into two groups, one with a high population level (POP = 1) and one with a low population level (POP = 0), based on the median level of urban population size. Separate empirical tests are conducted, and the findings are displayed in columns (1) and (2) of Table 12. Column (1) shows that the digital economy’s regression coefficient is −0.1878 in the sample group with a high population size. Column (2) suggests that in cities with low population levels, the digital economy’s regression coefficient is 7.4492, and it is significant at 1%, implying that the carbon emission intensity with highly populated cities might be decreased through the digital economy. As the digital economy develops, cities with small populations will experience a significant increase in carbon emission intensity, and the degree of change will be greater than in cities with a high population. The difference between the two groups of coefficients was tested, and the value of Chi2 was 33.96, which was significant at a 1% statistical level. The disparity could be explained by the fact that in major cities, the development of digital technology and e-commerce is based on wealthier markets, resulting in more active economic activities and a more noticeable decarbonization benefit of the digital economy. However, the digital economy is highly susceptible to carbon emissions in small cities with constrained resource allocation and market circumstances.

#### 5.5.3. Heterogeneity of Economic

The article groups the research samples and performs statistical tests based on the heterogeneity of economic levels. The samples are divided into a high economic development level group (LGDP = 1 group) and a low economic development level group (LGDP = 0 group) according to the median economic level, and the empirical tests are conducted respectively. The results are displayed in Table 12’s columns (3) and (4). Column (3) shows that the digital economy’s regression coefficient is 2.1807 in the sample group with a high economic development level, and column (4) indicates that the digital economy’s regression coefficient is 4.8340 in the sample group with a low economic development level, and both pass the 1% level significance test. The difference between the two groups of coefficients is tested, and the Chi2 value is 5.70, which is significant at 5%. According to the findings, when the level of the digital economy increases, the intensity of carbon emission intensity increases dramatically regardless of the regional economic development level.

## 6. Conclusions and Countermeasures

### 6.1. Conclusions

This article investigates the relationship between the digital economy and urban carbon emission intensity using panel data of Chinese cities to evaluate the digital economy’s growth level. The carbon emission intensity of major cities in China shows a clear pattern of “high in the west and low in the east”. The carbon emission intensity of the eastern, central and western regions is significantly different, and the regional interior is relatively close. In China’s largest cities, the growth of the digital economy exhibits clear ladder distribution features, with regional agglomeration and relative stability in the spatial dimension. However, there are obvious differences between regions, but there is a trend of narrowing differences on the whole.

It is discovered that urban carbon emissions can be directly decreased by the digital economy. According to the statistical yearbook data, the average value of urban carbon emission intensity is 0.4686, and it will decrease by about 0.3932 (0.8392 × 0.4686) for each 1% increase in the digital economy. The regression coefficient of the square term of the digital economy to carbon emissions is −1.2471, which is significant at the statistical level of 5%. Urban carbon emissions are affected by the digital economy in an inverted U-shaped pattern. An industrial structure rationalization can adjust the way the digital economy affects carbon emissions. Urban carbon emissions can rapidly increase in cities with inadequate industrial structure optimization because the digital economy is on the upswing. With other conditions unchanged, the average difference in carbon emission intensity between regions with the extremely high digital economy (DIG = 1 group) and extremely low digital economy (DIG = 0 group) is 2.4531 (5.235 × 0.4686). The primary source of both energy consumption and carbon emissions is industry. Enterprises promote green technology innovation and clean production, and the government implementing environmental regulation means it is important to achieve “double carbon” goals. According to the empirical test, the digital economy may effectively reduce carbon emissions through environmental regulation and green technology innovation. The regression coefficient of green technology innovation to carbon emissions is 0.0229, which is significant at the statistical level of 0.1. By fostering innovation in green technologies, the digital economy can lower the intensity of carbon emissions. The regression coefficient of environmental regulation on carbon emissions is 0.5438 and is at the 1% significant level. By reducing industrial waste emissions, the digital economy contributes to a reduction in the intensity of carbon emissions.

The digital economy’s decrease in urban carbon emissions has a clear regional variation, urban scale heterogeneity, and economic development level heterogeneity. The regression coefficients of the impact of the digital economy on carbon emissions in the eastern, central and western regions of China are −0.6258, −1.2342 and −1.3356, respectively, indicating that the digital economy has the effect of reducing carbon emissions in different regions. In densely populated cities, the digital economy can decrease the intensity of carbon emissions. For cities with low populations, the intensity of carbon emissions will considerably increase as the digital economy develops, and the degree of change is better than that of cities with a large population. The intensity of carbon emissions will dramatically grow with the expansion of the digital economy, regardless of whether the city has a high or low level of economic development. The above conclusions are still basically supported by the robustness test of the “Broadband China” policy impact, constructing lagged variables, changing the calculation method of the digital economy, replacing the measurement index of carbon emission intensity, and excluding provincial capitals and municipalities.

### 6.2. Countermeasures

In the context of “double carbon”, any country must strike a balance between economic development and environmental degradation. How to improve the development ability of the digital economy, restrain fossil energy consumption and carbon and nitrogen oxides emissions, and activate the digital economy’s ability in reducing carbon emissions has emerged as the essential component of economic transformation and development. This paper presents the following recommendations based on theoretical mechanism demonstration and empirical model tests, to strengthen the crucial role of the digital economy in reducing urban carbon emissions.

First, the government needs to enact laws and regulations to regulate individuals and organizations, develop the digital economy steadily and strategically, and formulate digital development regulations based on local conditions. Implementing the dynamic and differentiated regional digital industry development strategy and promoting the organic integration of the digital economy and the real economy will truly make the digital economy a key initiative to effectively reduce regional development imbalance. The second is to properly open up the market and provide an environment for technological investment to strengthen green technology innovation, thus reducing the intensity of energy consumption. By creating an atmosphere that encourages innovation, an inclusive environment for innovation and a guaranteed source of capital, the confidence and investment of enterprises in investing in green production technologies will be enhanced, fundamentally promoting the output of green technology innovation and enhancing clean technology research and development and green production patterns of enterprises. The third is to accelerate the development of digital architecture and expand and guarantee the scope of the role of data, technologies and other new factors of production and labor, capital and other factors of integration. By using digital network technology to integrate the allocation and flow of multiple production factors in the enterprise to reduce the cost of innovation. It is also vital to accelerate the diffusion of green innovation results, innovate resource-consuming traditional technologies, and so lower the intensity of carbon emissions. The fourth is to develop human capital to gain more innovation through rich expertise and experiential skills. In addition, the existing business needs to be reformed to improve overall efficiency and productivity, thereby saving resources and promoting the ultimate target of green development. The promotion of green innovation output by the digital economy is also a long-term process of dynamic structural complexity, and it does not just depend on one aspect of the digital economy. It is necessary to accurately judge the role and effect of the digital economy on regional green innovation output to achieve the dual goals of green economic development and carbon emission reduction. The fifth is to encourage industrial structure adjustment and realize the emission reduction effect of rationalization of industrial structure. In addition to facilitating the digital transformation of resource-based businesses, the digital economy also fosters the emergence and expansion of new industries and directs the coordinated growth of the industrial structure, which lowers carbon emissions. Industrial enterprises, as one of the core subjects to achieve the target of “double carbon”, can reduce energy consumption and achieve high-quality sustainable stable economic operations by actively participating in carbon trading, promoting low-carbon transformation and adopting green production.

### 6.3. Innovation and Deficiency

This paper may make the following innovations and contributions: First, based on existing literature research and economic theories, we propose the theoretical hypothesis that the digital economy affects carbon emissions. A total of 282 Chinese cities’ digital economic indices and the intensity of carbon emissions are measured by designing an index method. The range of potential theoretical applications of carbon emission reductions due to the digital economy has been expanded and improved. Second, the role of rationalization of industrial structure in the digital economy as it relates to carbon emissions is investigated, proposing the dual paths of green technology innovation and environmental regulation of the digital economy affecting carbon emissions, which offers a possible path choice for the digital economy’s carbon reduction effect. Third, the effects of the digital economy on carbon emissions have regional heterogeneity, population size heterogeneity, and economic development heterogeneity in China. Localized strategies are suggested for stakeholders at different levels to make rational decisions about carbon emission reduction and digital economic growth initiatives.

The research deficiencies may exist: Firstly, this paper discusses the spatial evolution characteristics of the digital economy index and carbon emissions based on urban panel data of China. To evaluate the spillover effect of the digital economy and the space-time transfer of carbon emissions, the possible spatial effects of the digital economy on carbon emissions can be subsequently studied. Secondly, the paper studies the impact of the digital economy on carbon emissions from the urban level. Subsequent studies can be discussed from the perspective of micro-enterprise entities.

## Figures and Tables

**Figure 1 ijerph-20-04931-f001:**
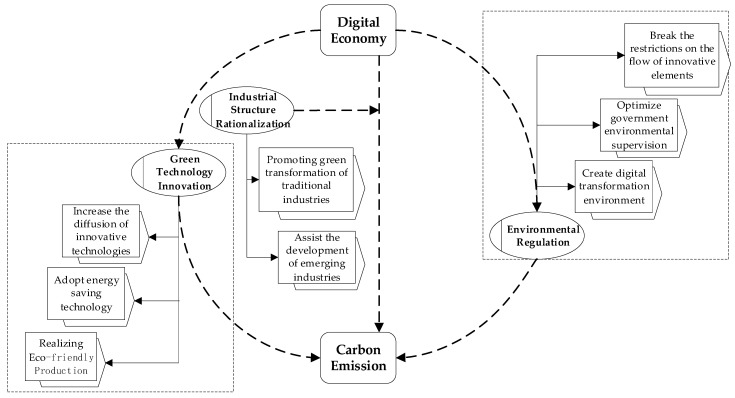
Theoretical framework diagram of the digital economy affecting carbon emissions.

**Figure 2 ijerph-20-04931-f002:**
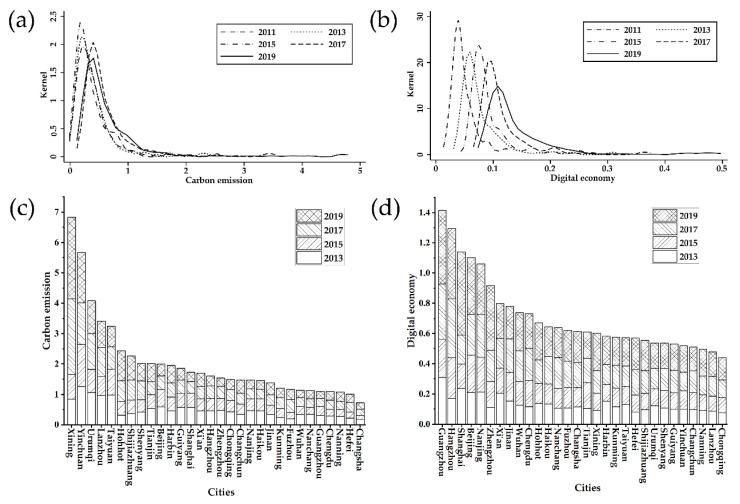
Spatial and temporal distribution of carbon emission intensity and digital economy. (**a**) The kernel density superposition of carbon emission intensity; (**b**) The kernel density superposition of digital economy; (**c**) The level of carbon emission intensity in China’s provincial capital cities and municipalities; (**d**) The development level of the digital economy in China’s provincial capital cities and municipalities.

**Table 1 ijerph-20-04931-t001:** Evaluation index system of comprehensive development level of the digital economy.

Primary Index	Secondary Index	Index Measure
Comprehensive development level of the digital economy	Internet infrastructure	Number of Internet users per 100 people (households)
	ICT application	Number of mobile phones per 100 people (units)
	Input of digital technical talents	Computer and software employees (%)
	Industry digitalization	Per capita postal business volume (RMB)
	Internet-related output	Total telecom business per capita (RMB)
	Digital industrialization	Digital inclusive finance index

**Table 2 ijerph-20-04931-t002:** Descriptive statistics of variables.

Variable Type	Variable Name	Symbol	Mean	Sd	Min	Max
Explained variable	Carbon dioxide emission	COE	0.4686	0.4215	0.0588	2.7973
Explanatory variable	Digital economy	DIG	0.0940	0.0491	0.0268	0.3239
Regulated variable	Rationalize the structure of production	INR	0.2740	0.1948	0.0040	0.8195
Mediator variable	Green technology innovation	GTI	4.2644	1.7804	0	8.7601
	Environmental regulation	ENV	0.0013	0.0889	−0.1086	0.4208
Control variable	Fixed capital investment stock	FAS	17.645	0.8713	15.7064	19.7441
	Industrial structures	IND	0.4697	0.1045	0.194	0.731
	Government scale	GOV	0.0788	0.0277	0.0342	0.1820
	Educational level	EDU	0.0346	0.0171	0.0133	0.1097
	Marketization level	MAR	11.6597	2.1835	6.9768	16.8957
	Openness to outside world	OPEN	0.1742	0.2726	0.0016	1.5745

**Table 3 ijerph-20-04931-t003:** Effect of the digital economy on carbon emissions.

	(1)	(2)	(3)	(4)	(5)
	COE	COE	COE	COE	COE
DIG	−1.0031 ***	−0.8392 ***			
	(−2.9701)	(−2.6202)			
DIGPF			−1.2471 **		
			(−2.0473)		
BRC				−0.0738 **	−0.0669 **
				(−2.1823)	(−1.9824)
FAS		−0.3640 ***	−0.3680 ***		−0.3569 ***
		(−3.8125)	(−3.8575)		(−3.7724)
IND		−0.5079	−0.5101		−0.5412
		(−1.0809)	(−1.0862)		(−1.1562)
GOV		1.1501	1.1208		1.2884
		(1.1669)	(1.1374)		(1.2801)
EDU		5.0974 **	5.0943 **		4.7324 **
		(2.5437)	(2.5482)		(2.3213)
MAR		−0.0014	−0.0008		−0.0003
		(−0.0757)	(−0.0451)		(−0.0171)
OPEN		0.2773 ***	0.2899 ***		0.3170 ***
		(3.6514)	(3.5500)		(4.0646)
CONS	0.4736 ***	6.6649 ***	6.6927 ***	0.7114	7.0581 ***
	(22.2893)	(4.0403)	(4.0683)	(22.9342)	(3.9602)
N	2538	2538	2538	2538	2538
Adj. R^2^	0.171	0.218	0.223	0.169	0.218

Note: Z values are in brackets. “**” and “***” are significant at the statistical level of 5% and 1% respectively.

**Table 4 ijerph-20-04931-t004:** Endogenous test.

	(1)		(2)
	COE		COE
L.DIG	−1.1493 **	L.DIG	−1.2476 **
	(−2.2853)		(−2.5124)
FAS	−0.3976 ***	L.FAS	−0.3135 ***
	(−3.6145)		(−2.6735)
IND	−0.5202	L.IND	−0.2480
	(−1.2502)		(−0.4498)
GOV	0.8048	L.GOV	−0.5575
	(0.8515)		(−0.5974)
EDU	5.5832 ***	L.EDU	5.5283 **
	(2.6851)		(2.4489)
MAR	−0.0142	L.MAR	−0.0138
	(−0.7898)		(−0.4612)
OPEN	0.2862 ***	L.OPEN	0.2880 ***
	(3.2201)		(3.1308)
CONS	7.4156 ***	_CONS	5.8959 ***
	(3.8297)		(2.8603)
N	2256	*N*	2256
Adj. R^2^	0.233	adj. R^2^	0.211

Note: Z values are in brackets. “**” and “***” are significant at the statistical level of 5% and 1% respectively.

**Table 5 ijerph-20-04931-t005:** Robustness test of changing the digital economy.

	(1)	(2)	(3)
	COE	COE	COE
DIG2	−0.0392		
	(−1.0955)		
DIG2PF		−0.0361 ***	
		(−3.5026)	
L. DIG2			−0.0676 **
			(−2.0085)
FAS	−0.2309 ***	−0.2599 ***	−0.2620 ***
	(−3.4594)	(−3.6354)	(−3.4233)
IND	−0.1073	−0.0773	−0.1776
	(−0.4838)	(−0.3383)	(−0.7696)
GOV	0.3683	0.4024	0.1586
	(0.6587)	(0.7238)	(0.2759)
EDU	6.3164 ***	6.2501 ***	6.5390 ***
	(3.6374)	(3.6201)	(3.4750)
MAR	−0.0008	0.0010	−0.0080
	(−0.0508)	(0.0671)	(−0.5069)
OPEN	0.3300 ***	0.2707 ***	0.3222 ***
	(3.8305)	(3.6509)	(3.0893)
CONS	4.1266 ***	4.6398 ***	4.7484 ***
	(3.6289)	(3.8214)	(3.5748)
N	2538	2538	2256
Adj. R^2^	0.308	0.317	0.331

Note: Z values are in brackets. “**” and “***” are significant at the statistical level of 5% and 1% respectively.

**Table 6 ijerph-20-04931-t006:** Robustness test of changing carbon emissions.

	(1)	(2)	(3)
	COE2	COE2	COE2
DIG	−1.3062 ***		
	(−3.1207)		
DIGPF		−2.0543 **	
		(−2.0770)	
L. DIG			−1.3560 **
			(−1.9834)
FAS	0.3919 ***	0.3847 ***	0.4413 ***
	(3.1596)	(3.1084)	(3.2087)
IND	1.0748 ***	1.0726 ***	0.8994 **
	(2.6176)	(2.6134)	(2.1453)
GOV	−0.4416	−0.4928	−0.9646
	(−0.4633)	(−0.5187)	(−0.9543)
EDU	1.7244	1.7233	2.0825
	(0.5822)	(0.5838)	(0.6556)
MAR	−0.0213	−0.0203	−0.0382
	(−0.6691)	(−0.6391)	(−1.1225)
OPEN	0.4174 ***	0.4340 ***	0.4408 ***
	(4.3725)	(4.0984)	(4.1134)
CONS	−1.1818	−1.1231	−1.7323
	(−0.5562)	(−0.5298)	(−0.7163)
N	2538	2538	2256
Adj. R^2^	0.606	0.607	0.604

Note: Z values are in brackets. “**” and “***” are significant at the statistical level of 5% and 1% respectively.

**Table 7 ijerph-20-04931-t007:** Robustness test excluding provincial capitals and municipalities.

	(1)	(2)	(3)
	COE	COE	COE
DIG	−0.8424 *		
	(−1.7681)		
DIGPF		−4.2181 ***	
		(−3.1485)	
L.DIG			−1.3221 **
			(−2.5531)
FAS	−0.3367 ***	−0.3507 ***	−0.3718 ***
	(−4.9965)	(−5.1466)	(−4.6821)
IND	0.1023	0.1325	0.0569
	(0.4760)	(0.6157)	(0.2585)
GOV	0.2320	0.2527	−0.0753
	(0.3770)	(0.4107)	(−0.1178)
EDU	5.6942 ***	5.6594 ***	6.0611 ***
	(3.1737)	(3.1649)	(3.1130)
MAR	−0.0003	−0.0004	−0.0043
	(−0.0215)	(−0.0234)	(−0.2633)
OPEN	0.4134 ***	0.3825 ***	0.4275 ***
	(3.8987)	(3.8297)	(3.3651)
CONS	5.8343 ***	6.0323 ***	6.5330 ***
	(5.0715)	(5.2128)	(4.7319)
N	2268	2268	2016
Adj. R^2^	0.342	0.345	0.365

Note: Z values are in brackets. “*”, “**” and “***” are significant at the statistical level of 10%, 5% and 1% respectively.

**Table 8 ijerph-20-04931-t008:** Inspection of the regulating effect of rationalization of industrial structure.

	(1)	(2)
	COE	
DIG	1.9094 ***	(7.0493)
INR	−0.4595 ***	(−9.6222)
DIG_INR	3.3256 ***	(6.8792)
FAS	−0.2480 ***	(−16.4067)
IND	0.3096 ***	(2.7094)
GOV	0.6560	(1.6024)
EDU	−2.5972 ***	(−3.4350)
MAR	0.0090 *	(1.7123)
OPEN	−0.1551 ***	(−3.7113)
CONS	4.5933 ***	(16.9617)
Fixed time	Yes	
Individual fixation	Yes	
Joint inspection	5.2350 ***	
N	2538	
Adj. R^2^	0.143	

Note: Z values are in brackets. “*” and “***” are significant at the statistical level of 10% and 1% respectively.

**Table 9 ijerph-20-04931-t009:** The intermediary effect test of green technology innovation and environmental regulation.

	(1)	(2)	(3)	(4)	(5)
	COE	GTI	COE	ENV	COE
DIG	−0.8442 ***	0.6149 *	−0.8582 ***	−0.2734 ***	−0.8354 **
	(−2.6335)	(1.9225)	(−2.6571)	(−3.4791)	(−2.1093)
GTI			0.0229 *		
			(1.8744)		
ENV					0.5438 ***
					(3.7887)
FAS	−0.3612 ***	0.6137 ***	−0.3752 ***	−0.0073	−0.2326 ***
	(−3.7873)	(5.3693)	(−3.8757)	(−0.5327)	(−3.5110)
IND	−0.5173	1.2832 ***	−0.5467	−0.1113 **	−0.0264
	(−1.1021)	(3.0411)	(−1.1673)	(−2.4418)	(−0.1174)
GOV	1.1656	0.9040	1.1449	−0.2100 *	0.5006
	(1.1832)	(0.8133)	(1.1684)	(−1.8074)	(0.8839)
EDU	5.1045 **	3.9174	5.0149 **	−0.4539	6.5656 ***
	(2.5466)	(1.1822)	(2.5165)	(−1.5590)	(3.7861)
MAR	−0.0014	−0.0384	−0.0005	−0.0013	−0.0003
	(−0.0748)	(−1.0660)	(−0.0275)	(−0.2952)	(−0.0190)
OPEN	0.2765 ***	0.1706	0.2726 ***	−0.0118	0.3087 ***
	(3.6415)	(1.3818)	(3.5849)	(−0.6910)	(3.8818)
CONS	6.6205 ***	−7.7513 ***	6.7980 ***	0.2422	4.1568 ***
	(4.0172)	(−3.9230)	(4.0888)	(0.9983)	(3.6555)
N	2538	2538	2538	2538	2538
Adj. R^2^	0.219	0.685	0.219	0.014	0.325

Note: Z values are in brackets. “*”, “**” and “***” are significant at the statistical level of 10%, 5% and 1% respectively.

**Table 10 ijerph-20-04931-t010:** Division of regions.

Zone	Province (City, District)
East	Beijing, Tianjin, Hebei, Liaoning, Shanghai, Jiangsu, Zhejiang, Fujian, Shandong, Guangdong, and Hainan
Central	Shanxi, Jilin, Heilongjiang, Anhui, Jiangxi, Henan, Hubei, and Hunan
West	Inner Mongolia, Guangxi, Chongqing, Sichuan, Guizhou, Yunnan, Shaanxi, Gansu, Qinghai, Ningxia, and Xinjiang

**Table 11 ijerph-20-04931-t011:** Regional heterogeneity in east, central and west China.

	(1)	(2)	(3)
	COE	COE	COE
DIG	−0.6258 ***	−1.2342 ***	−1.3356
	(−2.7282)	(−3.1133)	(−0.6775)
FAS	−0.2098 *	−0.2557 **	−0.5320 **
	(−1.9355)	(−2.5818)	(−2.1357)
IND	−1.2911 **	0.1196	0.5794
	(−2.4148)	(0.4934)	(1.1735)
GOV	1.4047	0.5595	1.1423
	(0.9460)	(0.8829)	(0.8459)
EDU	5.7893	1.9562	7.1565 **
	(1.2929)	(1.2530)	(2.1201)
MAR	−0.0287	0.0091	−0.0035
	(−1.3062)	(0.4069)	(−0.0928)
OPEN	0.4299 ***	0.0293	0.2837
	(3.2345)	(0.5193)	(1.0872)
CONS	4.6256 **	4.5021 ***	8.8072 **
	(2.1934)	(2.6879)	(2.0865)
N	900	891	747
Adj. R^2^	0.277	0.348	0.276

Note: Z values are in brackets. “*”, “**” and “***” are significant at the statistical level of 10%, 5% and 1% respectively.

**Table 12 ijerph-20-04931-t012:** Test of population heterogeneity and economic development level heterogeneity.

	(1)	(2)	(3)	(4)
	COE	COE	COE	COE
DIG	−0.1878	7.4492 ***	2.1807 ***	4.8340 ***
	(−1.2370)	(11.9125)	(6.5648)	(9.8067)
FAS	0.0487 ***	−0.4760 ***	−0.3356 ***	−0.1819 ***
	(3.4900)	(−13.8170)	(−14.5487)	(−8.2083)
IND	−0.3138 ***	0.7631 ***	0.2639	−0.1228
	(−3.2654)	(4.2443)	(1.4014)	(−0.8328)
GOV	−0.4048	0.3274	2.9934 ***	0.3025
	(−1.3758)	(0.4670)	(4.5469)	(0.6011)
EDU	0.0850	−4.6889 ***	−16.2777 ***	−1.4769 **
	(0.1227)	(−4.3108)	(−5.9377)	(−1.9921)
MAR	0.0112 ***	0.0033	0.0085	0.0090
	(3.1958)	(0.3352)	(1.0065)	(1.3361)
OPEN	0.1028 ***	−0.3635 ***	−0.1797 ***	−0.1529 *
	(3.6282)	(−4.5136)	(−3.3150)	(−1.7289)
CONS	−0.4847 *	7.9189 ***	6.2811 ***	3.2280 ***
	(−1.8791)	(13.8086)	(13.7872)	(8.9435)
Chi2	33.96 ***		5.70 **	
N	1269	1269	1269	1269
Adj. R^2^	0.059	0.171	0.166	0.115

Note: Z values are in brackets. “*”, “**” and “***” are significant at the statistical level of 10%, 5% and 1% respectively.

## Data Availability

Not applicable.

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
