# Peer review of "Research on the Effect of Digital Economy on Carbon Emissions under the Background of “Double Carbon”"

_ijerph, 2023, doi:10.3390/ijerph20064931_

Round 1

Reviewer 1 Report

Dear authors,

Thank you very much for sending your paper to the journal. The following issues should be fixed in the next version:

1-      What is the research gap and research question?

2-      The theoretical issues and prior studies should be up to date by the following papers:

-How Does the Digital Economy Affect Carbon Emission Efficiency? Evidence from Energy Consumption and Industrial Value Chain

-The Effect of CO2 Gas Emissions on the Market Value, Price and Shares Returns

-Effects of the Digital Economy on Carbon Emissions in China: A Spatial Durbin Econometric Analysis

-impact of digital economy development on carbon emission intensity in the Beijing-Tianjin-Hebei region: a mechanism analysis based on industrial structure optimization and green innovation

-The effect of knowledge management on intellectual capital, social capital, and firm innovation

3-The implications of the study should be highlighted in the paper

Author Response

Response to Reviewer 1 Comments

Paper ID:ijerph-2208002

Dear Editors and Reviewers:

First of all, we sincerely thank you for your time and constructive comments on our manuscript entitled “Research on the Effect of Digital Economy on Carbon Emissions under the Background of "Double Carbon"” (ID: ijerph-2208002). Those comments are all valuable and very helpful for revising and improving our paper, and they are important guidelines for our research. We have carefully studied these comments and revised them, and we hope they will be approved by you. The revised parts are marked in red in the paper. The main corrections in the paper and the point-to-point responses to the editors' comments are as follows.

Reviewer 1

Comments and Suggestions for Authors

Thank you very much for sending your paper to the journal. The following issues should be fixed in the next version:

Point 1: What is the research gap and research question?

Response 1: Thank you for your opinions. To improve and perfect this paper, we have completed research gap in lines 212 to 218 on page 5 emphatically. We highlighted the research question in lines 218 to 223 on page 5. We also stressed our contributions in lines 223 to 237 on page 5.

Point 2: The theoretical issues and prior studies should be up to date by the following papers:

-How Does the Digital Economy Affect Carbon Emission Efficiency? Evidence from Energy Consumption and Industrial Value Chain.

-The Effect of CO2 Gas Emissions on the Market Value, Price and Shares Returns.

-Effects of the Digital Economy on Carbon Emissions in China: A Spatial Durbin Econometric Analysis.

-impact of digital economy development on carbon emission intensity in the Beijing-Tianjin-Hebei region: a mechanism analysis based on industrial structure optimization and green innovation.

-The effect of knowledge management on intellectual capital, social capital, and firm innovation.

Response 2: Thank you for your suggestion. We have updated the prior studies of the article that mentioned the link between the digital economy and carbon emission in this study by replacing references [1] Lyu K, Yang S, Zheng K, Zhang Y. How does the digital economy affect carbon emission efficiency? Evidence from energy consumption and industrial value chain. Energies, 2023, 16(2): 761. [2] Chang X, Li J. Effects of the digital economy on carbon emissions in China: A Spatial Durbin econometric analysis. Sustainability, 2022, 14(24): 16624.

[3] Salehi M, Fahimifard S H, Zimon G, Bujak A, Sadowski A. The effect of CO2 gas emissions on the market value, price and shares returns. Energies, 2022, 15(23): 9221. in lines 31 to 34 on page 1.

About the theoretical issues, we replaced the prior reference [51] Sun Y, Hu Z. Upgrading digital economy industry and improving urban environmental quality. Statistics and Decision, 2021, 37, 91-95. with a new reference [51] Gu R, Li C, Yang Y, Zhang J, Liu K. Impact of digital economy development on carbon emission intensity in the Beijing-Tianjin-Hebei region: a mechanism analysis based on industrial structure optimization and green innovation. Environ-mental Science and Pollution Research, 2023. doi: 10.1007/s11356-023-25140-3. in lines 314 to 316 on page 7.

What is more, we replaced [54] Han L, Chen S, Liang L. Digital economy, innovation environment and urban innovation capability. Scientific Research Management, 2021, 42, 35. with a new reference [54] Salehi M, Fahimi M A, Zimon G, Homayoun S. The effect of knowledge management on intellectual capital, social capital, and firm innovation. Journal of Facilities Management, 2022, 20(5): 732-748. in lines 335 to 336 on page 7.

Point 3: The implications of the study should be highlighted in the paper.

Response 3: We think this is an excellent suggestion. The implications of the study are discussed in the conclusion section in lines 822 to 828, lines 830 to 833, lines 838 to 844, lines 845 to 851, lines 854 to 863 on page 23.

Thank you again for your constructive comments that help us a lot to improve the paper.

Reviewer 2 Report

The paper is relevant and topical. The presentation and flow are also good. It is supported by the research hypothesis and design. Extensive statistical analysis is needed. Plagiarism is also high

Author Response

Response to Reviewer 2 Comments

Paper ID:ijerph-2208002

Dear Editors and Reviewers:

First of all, we sincerely thank you for your time and constructive comments on our manuscript entitled “Research on the Effect of Digital Economy on Carbon Emissions under the Background of "Double Carbon"” (ID: ijerph-2208002). Those comments are all valuable and very helpful for revising and improving our paper, and they are important guidelines for our research. We have carefully studied these comments and revised them, and we hope they will be approved by you. The revised parts are marked in red in the paper. The main corrections in the paper and the point-to-point responses to the editors' comments are as follows.

Reviewer 2

Comments and Suggestions for Authors

Thank you very much for sending your paper to the journal. The following issues should be fixed in the next version:

Point 1: Is the research design appropriate? - Can be improved.

Response 1: Thank you for your recommendation, we think this is an excellent suggestion. We are so sorry that we missed the literature review in the prior study. Now we completed research gap, research question and contributions in lines 212 to 237 on page 5.

Point 2: Are the methods adequately described? - Must be improved.

Response 2: Thank you for your opinions. In the model setting part, the coefficients of each model are explained and we can find them in lines 373 to 375, lines 380 to 382 on page 8, and lines 388 to 390, lines 395 to 397, lines 402 to 406, lines 410 to 414, lines 417 to 421 on page 9.

Point 3: Are the results clearly presented? - Must be improved.

Response 3: We think this is an excellent suggestion. We assumed that the results are shortly presented and now we explained the results in detail in the conclusion section in lines 822 to 828, lines 830 to 833, lines 838 to 844, lines 845 to 851, lines 854 to 863 on page 23.

Point 4: Are the conclusions supported by the results? - Must be improved.

Response 4: Thank you for your thoughtful advice. We presumed that the conclusions are partly supported by the results, and we also supplemented it with one short paragraph in lines 903 to 911 on page 24.

Thank you again for your constructive comments that help us a lot to improve the paper.

Reviewer 3 Report

Dear Authors,

The paper is dealing with an interesting  topic of Carbon Emissions in the context of the economy's digital transformation. The problem presented in the article is current and important in terms of implementing sustainable energy solutions.

Assessing the content of the article as a whole, I conclude that, in general, the presentation of the issues raised in it does not raise objections. The division of the article into individual chapters is logical, and appropriate research methods were used. However, I believe that improvements can be made to the presented text.

Remarks:

          I suggest a broader review of the international literature. In many places, the references only refer to Chinese publications. This particularly applies to the results obtained in the study presented in the article and the comparison of these results with analogous studies carried out in other countries.

          I think that Authors' of the article omitted an important constraint regarding the impact of digital transformation on carbon emissions. The authors do not take into account the energy consumption of the IT infrastructure and data centres themselves. This consumption is constantly increasing. I suggest supplementing the text with this aspect.

Kind regards,

Author Response

Response to Reviewer 3 Comments

Paper ID:ijerph-2208002

Dear Editors and Reviewers:

First of all, we sincerely thank you for your time and constructive comments on our manuscript entitled “Research on the Effect of Digital Economy on Carbon Emissions under the Background of "Double Carbon"” (ID: ijerph-2208002). Those comments are all valuable and very helpful for revising and improving our paper, and they are important guidelines for our research. We have carefully studied these comments and revised them, and we hope they will be approved by you. The revised parts are marked in red in the paper. The main corrections in the paper and the point-to-point responses to the editors' comments are as follows.

Reviewer 3

Comments and Suggestions for Authors

Thank you very much for sending your paper to the journal. The following issues should be fixed in the next version:

Point 1: I suggest a broader review of the international literature. ln many places, the references only refer to Chinese publications. This particularly applies to the results obtained in the study presented in the article and the comparison of these results with analogous studies carried out in other countries.

Response 1: Thank you for your suggestion. We have updated the prior studies of the article that mentioned the link between the digital economy and carbon emission in this study by replacing references [1] Lyu K, Yang S, Zheng K, Zhang Y. How does the digital economy affect carbon emission efficiency? Evidence from energy consumption and industrial value chain. Energies, 2023, 16(2): 761. [2] Chang X, Li J. Effects of the digital economy on carbon emissions in China: A Spatial Durbin econometric analysis. Sustainability, 2022, 14(24): 16624.

[3] Salehi M, Fahimifard S H, Zimon G, Bujak A, Sadowski A. The effect of CO2 gas emissions on the market value, price and shares returns. Energies, 2022, 15(23): 9221. in lines 31 to 34 on page 1.

About the theoretical parts, we replaced the prior reference [51] Sun Y, Hu Z. Upgrading digital economy industry and improving urban environmental quality. Statistics and Decision, 2021, 37, 91-95. with a new reference [51] Gu R, Li C, Yang Y, Zhang J, Liu K. Impact of digital economy development on carbon emission intensity in the Bei-jing-Tianjin-Hebei region: a mechanism analysis based on industrial structure optimization and green innovation. Environ-mental Science and Pollution Research, 2023. doi: 10.1007/s11356-023-25140-3. in lines 314 to 316 on page 7.

What is more, we replaced [54] Han L, Chen S, Liang L. Digital economy, innovation environment and urban innovation capability. Scientific Research Management, 2021, 42, 35. with a new reference [54] Salehi M, Fahimi M A, Zimon G, Homayoun S. The effect of knowledge management on intellectual capital, social capital, and firm innovation. Journal of Facilities Management, 2022, 20(5): 732-748. in lines 335 to 336 on page 7.

Point 2: I think that Authors' of the article omitted an important constraint regarding the impact of digital transformation on carbon emissions.

Response 2: Thank you for your suggestion. We presented the carbon reduction effect of the enterprise's digital transformation in detail in the research hypothesis section in lines 271 to 279 on page 6.

Point 3: The authors do not take into account the energy consumption of the IT infrastructure and data centers themselves. This consumption is constantly increasing. I sug-gest supplementing the text with this aspect.

Response 3: Thank you for your opinions. We partly considered the impact of digital infrastructure construction on carbon emissions when we talked about the inverted U-shaped effect of the digital economy on carbon emissions in Chapter 5.2.2 on page 14. And thank you for your recommendation, we give detailed explanation on the energy consumption of the IT infrastructure and data centers in lines 605 to 607 on page 14 and lines 611 to 613 on page 15.

Thank you again for your constructive comments that help us a lot to improve the paper.

Round 2

Reviewer 1 Report

Dear author,

Thank you very much for sending your revised paper in the current version; you incorporated my comments in the current version.

Author Response

Dear Editors and Reviewers:

On behalf of my co-authors, we sincerely thank you for your time and constructive comments on our manuscript entitled “Research on the Effect of Digital Economy on Carbon Emissions under the Background of "Double Carbon"” (ID: ijerph-2208002). Those comments are all valuable and very helpful for revising and improving our paper, and they are important guidelines for our research. We have carefully studied these comments and revised them, and we hope they will be approved by you. The revised parts are marked in red in the paper. We have tried our best to revise our manuscript according to your comments.

We would like to express our great appreciation to you and reviewers for comments on our paper, and your constructive comments helped us a lot to improve the paper. Thank you and best regards.

Reviewer 2 Report

The author(s) has addressed the review suggestions to a large extent. However plagiarism is very high

Author Response

Dear Editors and Reviewers:

On behalf of my co-authors, we sincerely thank you for your time and constructive comments on our manuscript entitled “Research on the Effect of Digital Economy on Carbon Emissions under the Background of "Double Carbon"” (ID: ijerph-2208002). Those comments are all valuable and very helpful for revising and improving our paper, and they are important guidelines for our research. We have carefully studied these comments and revised them, and we hope they will be approved by you. The revised parts are marked in red in the paper. We have tried our best to revise our manuscript according to your comments.We would like to express our great appreciation to you and reviewers for comments on our paper, and your constructive comments helped us a lot to improve the paper. Thank you and best regards.
